# Experimental Examination of Solubility and Lipophilicity as Pharmaceutically Relevant Points of Novel Bioactive Hybrid Compounds

**DOI:** 10.3390/molecules27196504

**Published:** 2022-10-01

**Authors:** Angelica Sharapova, Marina Ol’khovich, Svetlana Blokhina, German L. Perlovich

**Affiliations:** G. A. Krestov Institute of Solution Chemistry, Russian Academy of Sciences, 1 Akademicheskaya Street, 153045 Ivanovo, Russia

**Keywords:** new antimycotic compounds, solubility, buffer solutions, partition coefficient, biphasic system, transfer, thermodynamics

## Abstract

The important physicochemical properties of three novel bioactive hybrid compounds with different groups (-CH_3_, -F and -Cl) were studied, including kinetic and thermodynamic solubility in pharmaceutically relevant solvents (buffer solutions and 1-octanol) as well as partition coefficient in system 1-octanol/buffer pH 7.4. The aqueous solubility of these chemicals is poor and ranged from 0.67 × 10^−4^ to 1.98 × 10^−3^ mol·L^−1^. The compounds studied are more soluble in the buffer pH 2.0, simulating the gastrointestinal tract environment (by an order of magnitude) than in the buffer pH 7.4 modelling plasma of blood. The solubility in 1-octanol is significantly higher; that is because of the specific interactions of the compounds with the solvent. The prediction solubility behaviour of the hybrid compounds using Hansen’s three-parameter approach showed acceptable results. The experimental solubility of potential drugs was successfully correlated by means of two commonly known equations: modified Apelblat and van’t Hoff. The temperature dependencies of partition coefficients of new hybrids in the model system 1-octanol/buffer pH 7.4 as a surrogate lipophilicity were measured by the shake flask method. It was found that compounds demonstrated a lipophilic nature and have optimal values of partition coefficients for oral absorption. Bioactive assay manifested that prepared compounds showed antifungal activities equal to or greater than fluconazole. In addition, the thermodynamic aspects of dissolution and partition processes have been examined. Bioactive assay manifested that prepared compounds showed antifungal activities equal to or greater than the reference drug.

## 1. Introduction

Invasive fungal infections have been the most frequent cause of death from infection around the world over the last 40 years, which is associated with the constant growth in the resistivity to the known antifungal drugs [1]. Most at risk are immunocompromised patients after transplantation, burn patients, people with diabetes, cancer, and immune diseases, as well as those who take broad-spectrum antibiotics. The causative agents of mycoses are also becoming more numerous. There are now more than 500 fungus species causing human diseases [2]. All this urges medicinal chemists to continue their search for rational approaches to the development of promising pharmacological agents with high activity, low toxicity, and multi-position mechanism of action. One of such directions is the concept of privileged structures [3]. Combining privileged structures when developing leader compounds increases the probability of finding novel highly active molecules hitting a variety of biotargets. The main approaches to creating such drug compounds are the design of new biologically active substances containing in their structure two or more pharmacophore groups or introduction of an additional bioactive group into the molecule of a known drug. Such compounds are called hybrids.

One of the privileged structures in the development of new antimycotic agents now is fluconazole, a known antifungal preparation, which is characterized by high activity, excellent safety profile, and favourable pharmacokinetic properties. It is used as a motif for creating hybrid compounds and appears to have a positive effect on bioactivity [4,5]. The thiazolo[4,5-d]pyrimidine backbone is also successfully applied in medicinal chemistry to design new therapeutic preparations due to its wide spectrum of pharmacological activity, including antibacterial, antiviral, anti-tumour, pain relieving, and anti-inflammatory actions [6].

The objects of this study are new hybrid compounds containing a combination of fluconazole and a thiazolo[4,5-d]pyrimidine fragment linked by a piperazine spacer [7]. The synthetic pathway of target hybrids is presented in Figure 1.

In the past, researchers used to rely only on biological activity when developing drug compounds but pharmacokinetic problems at the last stages of the process often turned their attempts to create new therapeutic agents into a failure. A successful drug combines biological activity and pharmaceutically relevant physicochemical properties, such as solubility, lipophilicity, stability, ionization capacity, etc. That is why at the first stages of screening of new bioactive agents, modern science evaluates their solubility and lipophilicity as the main pharmaceutical characteristics. Solubility in water is the key parameter of potential candidates for becoming drug compounds [8]. Poor solubility in water often leads to poor absorption, even if the permeation rate is high because the drug compound penetration through the intestine membrane is proportional to the gradient of the drug concentration between the lumen of the intestine and blood. Besides, high concentrations of poorly soluble drug compounds in the body may lead to crystallization and high toxicity. Lipophilicity is traditionally characterized by the partition coefficient logarithm in the 1-octanol/water system and has a significant effect both on the pharmacokinetic and pharmacodynamic behaviour, as well as toxicology [9]. Besides, this parameter is widely used in drug discovery as a descriptor for QSAR (quantitative structure − activity relationship) and QSPR (quantitative structure − property relationship) approaches [10]. Solubility and lipophilicity screening at the in vitro stage is important for speeding up the selection of the most promising candidates among new bioactive compounds, reducing the necessity to conduct expensive laboratory experiments and biological tests.

It should be said that there is no universal medium that could be used to predict the solubility of every drug compound or its behaviour during dissolution in vivo. Aqueous solubility of compounds at room temperature is usually employed as a model in pharmacopeial standards [11]. However, in most cases, the pH value and medium composition are important and may affect the drug solubility, for example, through ionization of functional groups [12]. That is why the solubility measured in water can be different from that in the gastro-intestinal tract, especially in case of lipophilic and poorly soluble drugs. In our work, the hydrochloric buffer pH 2.0 and phosphate buffer pH 7.4 imitating pH of the fasted stomach and blood plasma, respectively, were selected as the dissolution media for predicting clinically relevant properties of new hybrid compounds [9]. Besides, the solubility of the compounds under study was determined in 1-octanol, which quite well imitates the properties of the biological membrane phospholipids due to its amphiphilic nature (polar “heads” and flexible nonpolar “tails”) and ability to form hydrogen bonds. Solubility in 1-octanol is an indicator of how easily a drug will permeate the biological membranes, how strongly it will bind to its receptor, and how long it will remain in the body [13].

In view of all the above, the aim of this work is to study the dissolution and partition processes of three new potential antimycotic agents in model biological media, solving the following problems:−Evaluation of the kinetic solubility in buffer solutions of various acidity degrees; −Determination of the temperature dependence of equilibrium solubility in buffer solutions and 1-octanol;−Measurement of partition coefficients in the two-phase system of immiscible solvents—1-octanol/buffer pH 7.4—within the temperature range of 293.15–313.15 K;−Calculation of the thermodynamic parameters of dissolution and partition processes in the studied systems and identification of their driving forces.

The studies conducted will not only allow researchers to do a detailed analysis of the physicochemical and biopharmaceutical properties of new hybrid compounds but will also provide additional information about the structure–property relationship for the compounds of similar chemical nature studied by us earlier [14,15]. Such rational evaluation will promote the development of high quality and effective antifungal agents.

## 2. Results

### 2.1. Bioactive Assay

The in vitro antifungal screening results of synthesized compounds was evaluated against three fungal strains: *C. parapsilosis* ATCC 22019, *C**. utilis 84,* and *C. glabrata* 61L are summarized in Table 1.

According to the obtained results, it was concluded that all of hybrids studied effectively inhibited the growth of tested pathogens. The compounds I–III exhibited excellent or good antifungal activities, with MIC values ranging from 0.1 to 4 μg/mL. For *C. parapsilosis* ATCC 22019, compounds I–III showed 4-fold, 20-fold, and 8-fold advantages over fluconazole, respectively. Obviously, the most active in this test is the hybrid with a fluorine substituent. Whereas the growth of haploid fungi *C. glabrata* 61L is better inhibited by compounds I and III with CH_3_- and Cl- groups in the structure, MIC values are four times less than the reference value. With regards to the fungus *C. utilis* 84, compound II exceeds the activity of fluconazole by 2 times, and compound III is comparable to this known drug. Thus, it may be concluded that the application of the molecular hybridization approach improved the antifungal activity of the target molecules.

### 2.2. Kinetic Solubility

The kinetic (non-equilibrium) solubility is usually employed at the early stages of development of new bioactive chemical compounds. It is unclear whether data on kinetic solubility can be suitable and applicable to the development of a formulation and a dosage form, but this inner molecular solubility can be improved by modifying the chemical structure, which provides important information for design of structurally related compounds [16]. Besides, the data on kinetic solubility make it possible to determine the time required to reach the equilibrium in the solvent/solute system, which is the key factor in thermodynamic (equilibrium) solubility evaluation.

To obtain solubility kinetic dependences of the hybrid compounds, buffer solutions pH 2.0 and 7.4 simulating the gastric juice and blood plasma media, respectively, were used. It is in these media that the most intensive dissolution and absorption of a drug take place, depending on the introduction method. The kinetic curves reflecting the dependence of changes in the concentration of the initial substances on time are presented in Figure 2. Moreover, the experimental data are tabulated and presented in Appendix A.

It should be said that the kinetic curves of all the synthesized compounds are identical in their shape. Analysis of the obtained dependences allows us to obtain the following regularities of the dissolution kinetics of compounds I–III in aqueous solutions: (a) the solubility values in the hydrochloric buffer with pH 2.0 are much higher than those in the phosphate buffer solution with pH 7.4; (b) extreme solubility growth is observed in the first 4 h of the experiment and then the value rises; monotonically reaching a plateau (in the buffer solution with pH 7.4 the solution saturation takes less time ~300 min, whereas in the buffer pH 2.0, this process takes from 1000 to 2200 min; (c) depending on the substituent nature, the solubility in the buffer solution with pH 7.4 increases in the following series -Cl < -F < -CH_3_, and the trend is reversed for the solubility dependence on the time required for the system to reach an equilibrium. It is evident that the introduction of halogen atoms has a negative effect on the pharmacokinetic characteristics of the hybrid compounds. It is undoubtedly interesting to analyse the bottom phases extracted after the solubility experiment because amorphous substances can have higher apparent solubility that cannot be reached in toxicological studies. The solid phases of the studied compounds (carefully dried in advance) were characterized by the DSC and PXRD methods. As a result, the obtained PXRD patterns and DSC profiles of residual materials are completely identical to the results shown in Appendix A. The fact that the thermograms and diffraction patterns have no sharp peaks indicates that the compounds under study did not change their amorphous structure and are not capable of crystallization in aqueous solutions. Besides, the analysis done excludes the formation of saline or hydrated forms. Thus, it can be summarized that the measured kinetic solubility in the saturation section will also be the thermodynamic solubility of the hybrid compounds under consideration.

### 2.3. Equilibrium Solubility

The experimental solubility values of hybrids in the selected solvents within the temperature range of 293.15–313.15 K expressed in molarity are summarized in Table 2. The comparative analysis of the reference values of the solubility of compounds I–III at the standard temperature of 298.15 K is given as a histogram in Figure 3. 

As the obtained data show, all the studied compounds, as most organic substances, are characterized by a positive temperature coefficient, i.e., the solubility increases monotonically as the temperature goes up in all the solvents used.

An analysis of the data in Table 2 shows that the order of changes in the solubility values of the studied compounds, depending on the solvent nature, is expressed by the inequality: buffer pH 7.4 < buffer pH 2.0 < 1-octanol. The hybrid compounds under consideration are poorly soluble in the buffer solution pH 7.4 and have similar solubility values (*S*) changing at *T* = 298.15 K within the range (0.67–3.19) × 10^−4^ mol∙L^−1^ and increasing in the following order: III (chloro-) < I (methyl-) < II (fluoro-). At the same time, in the buffer pH 2.0, this sequence looks like: III (chloro-) < II (fluoro-) < I (methyl-). Furthermore, the *S* values range from 0.68 × 10^−3^ to 1.98 × 10^−3^ mol∙L^−1^, which is an order of magnitude higher than the solubility in the alkaline buffer solution. On the whole, it is evident that the studied compounds have limited solubility in aqueous media. This may be attributed to the high capacity of hydrogen bond formation between the solvent molecules, which leads to strong association of the water molecules and limits the dissolution of the rather bulky hybrid compounds in the buffer solutions.

The differences in the solubility of the compounds under study depending on the buffer solution pH value are most probably associated with molecule ionization. The apparent ionization constants of hybrids I–III refined using the *pDISOL-X* software (Rev 3.093) developed by Avdeef [17,18] are identical and equal to 8.1. The studied compounds are weak bases capable of binding a proton to the lone pair of electrons of the nitrogen atom in the piperazine heterocycle. The content of the ionized hybrid forms in the aqueous solution, depending on the medium pH, is presented in Figure 4. Theoretically, all the studied compounds are completely ionized in the acidic medium (buffer pH 2.0). When the pH value of the medium increases to 7.4, the content of charged particles goes down to 80%. A direct consequence of this process is the reduction in the solubility values in the phosphate buffer pH 7.4 for all the studied compounds. Figure 4 also shows the experimental points corresponding to the solubility of the studied compounds in the selected buffer solutions at *T* = 298.15 K. The obtained pH dependences are smooth asymptotically decreasing curves, which are typical for weak bases.

The investigated hybrid compounds are much better dissolved in amphiphilic 1-octanol than in the buffer solutions (Figure 3), which indicates the high lipophilicity of the substances and good permeation properties making their application in the form of ointments and creams favourable. The 1-octanol molecules with long hydrophobic tails of eight carbon atoms and hydrophilic hydroxyl groups can participate in van-der-Waals interactions and also form intermolecular hydrogen bonds with the solvent molecules. The presence of 10 H-acceptor sites (fluorine, chlorine, sulphur, nitrogen, and oxygen atoms) and the proton-donor hydroxyl group is responsible for the good potential hydrogen bonding capacity of these molecules. Thus, in the hybrid compound/1-octanol system, the determinant factor of the solubility value is the sum of nonspecific interactions and hydrogen bond energy. Since the acidic properties of 1-octanol are poorer than those of water, this results in the formation of weak hydrogen bond associations, which makes the drug solubility higher than in the aqueous solvents. It seems interesting that a similar inequality has been obtained for fluconazole and its hybrid compounds investigated by us earlier: S (1-octanol) > S (buffer solutions).

The solubility values of compounds I–III in the alcohol change as follows, depending on the radical nature: II (fluoro-, S = 5.67 × 10^−3^ mol∙L^−1^) < I (methyl-, S = 7.54 × 10^−3^ mol∙L^−1^) < III (chloro-, S = 1.23 × 10^−2^). Unfortunately, to unambiguously identify the ratio of the solubility values in the solvents used and the molecular structure of the compounds was not possible. This fact agrees with the fundamental postulate that the dissolution is a complex process that is not only dependent on structural factors but requires taking into consideration the solute and solvent physicochemical properties (shape and size of the molecules, polarity, ionization capacity, etc.) and ability of compounds to participate in various interactions (induced, dipole, and electrostatic interactions) [19].

### 2.4. Application of Hansen Solubility Parameters

The last few years have seen a successful application of Hansen’s three-parameter method for prediction of solubility of pharmaceutical substances in low-molecular weight solvents [20]. According to this approach, the solubility parameters of any chemical consist of three components—*δ_d_*, *δ_p_*, and *δ_h_*—characterizing the dispersion forces, polar interactions, and hydrogen bond formation, respectively [21]. The partial solubility parameters (*δ_d_*, *δ**_p_*, *δ**_h_*) are calculated by the following equations:*δ_d_* = *ΣF_di_/ΣV_i_*(1)
*δ_p_* = (*ΣF_pi_*^2^)^0.5^/*ΣV_i_*;(2)
*δ_h_* = (*ΣF_hi_*/*ΣV_i_*)^0.5^(3)
where *V_i_* denotes the contribution of the molar group *i* to the molar volume of the molecule and *F_di_*, *F_pi_*, and *E_hi_* are defined as contributions of dispersion forces, polarity and hydrogen bond structural group *i* to the cohesion energy. The values of these group components for compounds I–III are presented in Appendix A.

According to Hansen’s theory, the hydrogen bonding parameter includes all the energy contributions that are not described by the two other parameters. That is why the total parameter taking into account all types of interactions is found by the following expression: (4)$$\delta_{t}^{2}=\delta_{d}^{2}+\delta_{p}^{2}+\delta_{h}^{2}.$$

The difference in calculated parameters of solute (1) and solvent (2) was determined according to Equation (5) as the Euclidean distance or radius interaction:(5)$$\Delta \delta ={\left({\left(\delta_{d1}-\delta_{d2}\right)}^{2}+{\left(\delta_{p1}-\delta_{p2}\right)}^{2}+{\left(\delta_{h1}-\delta_{h2}\right)}^{2}\right)}^{0.5}\;$$

To calculate the partial solubility parameters, the group contribution parameters developed based only on pharmaceutical solid substances by Just et al. [22] were used. As the authors state, the set of new group contributions proposed by them ensures the higher overall predictive power for solubility experiments due to the introduction of new functional groups, compared to the established parameters used by van Krevelen/Hoftyzer [23] and others. The calculated values of the Hansen solubility partial parameters *δ_d_*, *δ_p_*, and *δ_h_* of the studied compounds and the solvents used are summarized in Table 3.

An analysis of the data in the Table 3 shows that the hybrid compounds under study have close solubility values. This fact is quite natural as the replacement of only one radical atom in the structure of the heterocyclic compound cannot significantly affect its physicochemical profile. The maximum values for all the compounds are found in the HSP dispersion component, which indicates the determinant role of these interactions in the dissolution process. Despite the presence of donor-acceptor groups in the structure of the compounds, the specific interaction component *δ**_h_* is more than 50% smaller than that of the nonspecific component. The smallest contribution to the total cohesion energy is made by the polar forces.

The main idea of compatibility between two substances is based on the golden rule of dissolution—like dissolves like. It means that if the parameters of a solute and a solvent coincide, the dissolution is effective. In our case, a good solvent for hybrid compounds is 1-octanol, the HSPs of which are close to the values of the parameters of the studied compounds. The polarity and especially the hydrogen bonding capacity of the buffer solutions are a lot different from the parameters of compounds I–III, which indicates poor compatibility of this type of compounds with aqueous media. The simplest and most convenient tool for evaluation of the miscibility degree is the Δ*δ**_t_* parameter. Greenhalgh et al. [24] determined the limits for Δ*δ**_t_*: if Δ*δ**_t_* < 7.0 MPa^0.5^, the components mix well, whereas Δ*δ**_t_* > 10.0 MPa^0.5^ indicates their incompatibility. Our calculations confirm good compatibility of the hybrid compounds with the alcohol (Δ*δ**_t_* < 3.0 MPa^0.5^) and very weak solubilizing effect of the buffer solutions (Δ*δ**_t_* > 24 MPa^0.5^). It should be said that the solubility behaviour predicted based on HSPs adequately reflects the experimental results obtained in this work.

### 2.5. Solubility Modeling

Two rather simple and well-established models to simulate the equilibrium solubility of compounds I–III in the selected solvents: the modified Apelblat equation and classical van’t Hoff equation were applied. Appendix A contain experimental and calculated solubility values of the hybrid compounds, as well as the coefficients of the equations and estimated parameters of the models (RMSD and RAD).

The comparison of fitting by two thermodynamic models in terms of the RAD for the whole of the experimental measurements graphically presented in Figure 5. It is obvious that for all the studied systems (except compound II in buffer pH 7.4), the modified Apelblat equation gives more accurate results. The maximum value of 100RAD is 0.26 for the Apelblat model and 0.30 for the van’t Hoff equation, whereas the average values are equal to 0.10 and 0.21, respectively.

There is a significant difference between the RMSD values in the systems with aqueous solutions and 1-octanol: the highest value for the buffer used 7.14 × 10^−8^ is found in the Apelblat model and the highest in the van’t Hoff model is 9.7510^−8^, whereas in the system with the alcohol, these values are equal: 2.03 × 10^−6^ and 2.03 × 10^−5^. The average RMSD values are equal to 1.85 × 10^−8^ for the buffer solutions and 1.68 × 10^−6^ for 1-octanol (Apelblat model), and, respectively, 2.98 × 10^−8^ and 8.36 × 10^−6^ (van’t Hoff equation). Thus, the comparison of the estimation parameters has shown that the fitting effect of the modified Apelblat model is relatively better. However, the presented data indicate that both equations give a satisfactory correlation’s results to the experimental data and can be used to predict solubility hybrid compounds of this class in different solvents using the interpolation technique.

### 2.6. Dissolution Thermodynamics

The dissolution of the new hybrids was characterized based on apparent thermodynamic parameters obtained from temperature dependences of solubility of the studied compounds in the selected solvents (Figure 6). As Figure 5 shows, in all systems studied, there is a good linear dependence of the experimental solubility values on temperature within the whole range (*R* ≥ 0.9994). The van’t Hoff enthalpy of dissolution can be expressed as the inclination of a tangent to a solubility curve plotted as ln*x* vs. (1/*T* − 1/*T_hm_*) by Equation (6): (6)$$\Delta H_{sol}^{o}=-R\left(\frac{\partial \ln x}{\partial \left(1/T-1/T_{hm}\right)}\right)=-R\cdot slope\;$$
where *R* = 8.314 J⋅K^−1^·mol^−1^ is the universal gas constant. *T_hm_* = 302.98 K denotes the mean harmonic temperature, which was calculated by Equation (7):(7)$$T_{hm}=\frac{n}{\sum_{i=1}^{n}\frac{1}{T_{i}}}\;$$
where *n* represented the number of experimental temperatures. The apparent standard Gibbs energy was evaluated by Equation (8):(8)$$\Delta G_{sol}^{o}=-RT_{hm}\times \mathrm{int}ersept\;$$

The apparent standard solution entropies were obtained from the well-known relationship:(9)$$\Delta S_{sol}^{o}=\frac{\Delta H_{sol}^{o}-\Delta G_{sol}^{o}}{T_{hm}}\;$$

The parameters of the empirical equation lnx = A − B(1/*T* − 1/*T_hm_*) are shown in Appendix A. The calculated apparent thermodynamic solubility functions of dissolution (Gibbs energy, enthalpy, and entropy) at standard temperature 298.15 K for hybrid compounds were collected in Table 4. 

An analysis of the presented data shows that the Gibbs energy of dissolution of the hybrid compounds in all the studied solvents is positive, which means the dissolution process is not spontaneous. The lowest values are found in the alcohol and the maximum ones in the buffer pH 7.4, which agrees with the solubility values obtained. The dissolution enthalpies of compounds I–III are also positive and, therefore, the dissolution process is endothermic. This indicates that the energy of the newly formed bond between the solute and solvent molecules is not enough to compensate for the energy needed to break the original association bond in the selected solvents. The entropy component of the Gibbs energy is negative in all the solvents studied. Such behaviour, as a rule, is explained by the hydrophobic effect typical of solutions of nonpolar organic compounds in aqueous media [25]. However, the nature, driving forces, and mechanism of these interactions still remain unclear. Frank and Evans [26] associate this phenomenon with the formation of clathrate-like structures around the nonpolar fragments of the solute molecule, which leads to a great decrease in the entropy. At the same time, Huyskens’ mobile order theory [27] states that hydrophobic effects are associated exclusively with the solvent self-association through intermolecular hydrogen bonds. This hypothesis is confirmed by the fact that the hydrophobic effect in water is a particular case of the more general solvophobic effect observed in various solvents where hydrogen bonds are formed, including alcohols. This is validated by the results of our study as the dissolution entropy values of the compounds in 1-octanol are also negative. It should be said that the solvophobic force of alcohols is much weaker than the hydrophobic interactions [28], which agrees with the data for compounds I and III: (1-octanol) > (buffer solutions). A compensation effect is observed for thermodynamic parameters of the studied compounds in the selected solvents (Appendix A). The phenomenon of enthalpy–entropy compensation consists in the reduction in the system configuration freedom and, hence, entropy reduction due to the stronger interactions between the molecules. In all the compounds, the dissolution process in the selected solvents is enthalpy-determined: $\Delta H_{sol}^{o}$ > *T*$\Delta S_{sol}^{o}$.

### 2.7. Partition in System 1-Octanol/Buffer pH 7.4 and Transfer Thermodynamics

The partition coefficients of the new hybrid compounds in the model system 1-octanol/buffer pH 7.4 as a lipophilicity surrogate were determined experimentally. Table 5 shows the concentrations of compounds I–III in the mutually saturated solvents used and partition coefficients log*P_o/b_* within the temperature range 293.15–313.15 K. The log*P_o/b_* are positive for all the compounds and increase with temperature growth, which indicates the shift of equilibrium to the organic phase imitating the membrane lipid layer. The obtained values of log*P_o/b_* > 1 demonstrate the hydrophobic nature of the studied compounds associated with their poor aqueous solubility. It is the lipophilic properties that allow the hybrids to move from the water phase to the less hydrophilic octanol phase, where the solute–solvent intermolecular interactions are more favourable. According to the results shown in Table 5, the order of changes in the partition coefficients log*P_o/_*_b_ at T = 298.15 K, depending on the substituent nature, is expressed as the inequality: Cl- (1.65) < F- (2.40) < CH_3_- (2.50). Compounds I and II are the most lipophilic in the series of the studied substances, whereas the introduction of a chlorine atom considerably lowers the partition coefficient (~0.8 log units). This fact agrees with the common concept that lipophilic groups are usually hydrocarbon or fluorine-containing radicals [29]. Thus, the studied compounds do not only obey Lipinski’s rule for “drug-like” chemicals: log*P_o/b_* < 5 [30] but also have the optimal partition coefficient values for oral absorption: 1 < log*P_o/b_* < 3 [31]. For drugs with such partition coefficients, there is a good balance between the solubility and permeability through passive diffusion and minimum metabolism due to the low binding capacity with enzymes. It should be noted that the results of biological tests (Table 2) are in agreement with the data on lipophilicity, namely, the hybrid compounds have the recommended values log*P_o/b_* and demonstrate significant antifungal activity.

The temperature variation of the partition coefficient provides enthalpic and entropic contributions to the free energy of transfer. The thermodynamic transfer functions for hybrids I–III in the 1-octanol/buffer pH 7.4 system were calculated by Equations (12)–(16) and are summarized in Table 6. The standard Gibbs energy values for all the compounds are negative. Therefore, the transfer from the aqueous medium to the organic phase within the studied temperature range is thermodynamically favourable and spontaneous. The standard transfer enthalpies of all the studied compounds are positive, which means the process is endothermic. Energy is consumed during the transfer to overcome the hydrophobic effects in water (destruction of the molecule solvate shell structure), destruction of the solvent–solute interactions in the aqueous medium and creation of a cavity in the organic phase where the solute molecules can enter.

The trend of transfer enthalpy increase in the series of the studied compounds correlates with growth in their lipophilicity. The maximum value of the transfer enthalpy for compound I indicates that the largest methyl group does not promote the transfer to the 1-octanol phase. This fact agrees with statement [32] that the larger the solute volume, the more solvent–solute bonds must be destroyed, and the more energy is consumed to create cavities. The high positive values of the transfer entropy of compounds I–III indicate that the transfer increases the system disordering.

## 3. Materials and Methods

### 3.1. Materials

The bidistilled water with electrical conductivity 2.1 μS cm^−1^ and pH 6.6 was used for preparation of buffer solutions. Phosphate buffer pH 7.4 (I = 0.15 mol·L^−1^) was made by combining the KHPO_4_ (9.1 g in 1 L) and NaH_2_PO_4_∙12H_2_O (23.6 g in 1 L) salts. For the preparation of the buffer solution pH 2.0 (I = 0.10 mol·L^−1^), 6.57 g of KCl was dissolved in water, 119.0 mL of 0.1 mol·L^−1^ hydrochloric acid was added, and the volume of the solution was adjusted to 1 L with water [33]. The final pH values were measured by pH meter FG2-Kit (Mettler Toledo, Greifensee, Switzerland). The device was calibrated with a two-point calibration using pH 4.00 and 9.22 solutions (accuracy ≥ 97%). All chemicals and reagents were used as obtained from the suppliers (without purification). The detailed information about all chemicals used in this work is given in Table 7.

### 3.2. Synthesis and Characterization

The methodology of synthesis of three novel antimycotic compounds as objects of the present investigation were synthesized by the methodology described in detail earlier [7]. In brief, a series of hybrid derivatives containing a piperazine linker between the triazole and pyrimidino[4,5-d]thiazol heterocycles was obtained by alkylation of the piperidine fragment imino group with haloalkyl. For that purpose, commercial haloalkyl Boc-piperazines: tert-butyl 4-(2-bromoethyl)piperazin-1-carboxylate and tert-butyl 4-(2-bromopropyl)piperazine-1-carboxylate were used, as well as the product of Boc-piperazine condensation with racemic epichlorohydrin produced by the method described in patent literature [34]. After removing the Boc-protection from the obtained products by trifluoroacetic acid in methylene chloride, the resulting piperazinyl thiazolo [4,5-d]pyrimidine derivatives were condensed by triazole oxirane in an alcohol with triethylamine and ammonium chloride to obtain target derivatives: 6-[3-[4-[2-(2,4-difluorophenyl)-2-hydroxy-3-(1*H*-1,2,4-triazol-1-yl)propyl]piperazin-1-yl]propyl]-3-(R-phenyl)-2-thioxo-thiazolo[4,5-d]pyrimidin-7-ones (R: CH_3_- (I); F- (II); Cl- (III)). The compounds were purified by column chromatography with silica gel in the ethylacetate:methanol (10:1) system. Afterward, the unambiguous structure and purity of the compounds under study was proved by a series of (^1^H, ^13^C) NMR experiments and LS-MS spectroscopy. NMR spectra were recorded on a Bruker Advance spectrometer (400, 600, 700 MHz and 151, 176 MHz).

6-[3-[4-[2-(2,4-difluorophenyl)-2-hydroxy-3-(1H-1,2,4-triazol-1-yl)propyl]piperazin-1-yl]propyl]-3-(p-tolyl)-2-thioxo-thiazolo[4,5-d]pyrimidin-7-one (I). Mix of 0.401 g (0.10 mmol) 6-(3-piperazin-1-ylpropyl)-3-(p-tolyl)-2-thioxo-thiazolo [4,5-d]pyrimidin-7-one (R^1^ = Me, R^2^ = H) and 0.33 g (0.10 mmol), metansulfonate compound, added 0.15 mL (0.15 mmol) triethyl amine, ammonium chloride on the tip of spatula in ethanol (20 mL) were added and heated for 13 h. After cooling the reaction mixture to room temperature, 10 mL water was added and the mixture was subjected to extraction by ethylacetate (3 × 30 mL). The combined organic phase was washed with a saturated NaCl solution (50 mL), dried over sodium sulphate, and evaporated. The yield was 0.26 g (41%) as light-yellow powder; C_30_H_32_F_2_N_8_O_2_S_2_; *m*/*z* 638.76. LCMS [M + 1]^+^ = 638.1^+1^.

^1^H NMR spectrum (Chloroform-d, δ, ppm, *J*/Hz): 2.26–2.76 (m, 12H, CH_3_, 5CH_2_), 3.06 (d, ^3^*J*_HH_ = 14.37, 1H, CH_2_), 3.73 (dd, ^3^
*J*_HH_ = 13.91, 8.47, 2H, CH_2_), 3.89–4.13 (m, 2H, CH_2_), 4.31 (t, ^3^
*J*_HH_ = 13.75, 1H, CH-OH), 4.45–4.61 (m, 2H, CH_2_), 6.75–6.91 (m, 2H, arom.), 7.23 (d, ^3^
*J*_HH_ = 8.13, 2H, arom.), 7.40 (d, ^3^
*J*_HH_ = 8.01, 2H, arom.), 7.49–7.62 (m, 1H, arom), 7.81 (d, ^3^
*J*_HH_ = 1.41, 1H, N=CH-N), 8.12 (m, 2H, N=CH-N).

6-[3-[4-[2-(2,4-difluorophenyl)-2-hydroxy-3-(1H-1,2,4-triazol-1-yl)propyl]piperazin-1-yl]propyl]-3-(4-fluorophenyl)-2-thioxo-thiazolo[4,5-d]pyrimidin-7-one (II) was prepared as a light-yellow powder in 36% yield (0.198 g) from 3-(4-fluorophenyl)-2-methyl-6-(3-piperazin-1-ylpropyl)-2-thioxo-thiazolo[4,5-d]pyrimidin-7-one (R^1^=F, R^2^=H) (0.42 g, 0.1 mmol) and compound 1 (0.33 g, 0.10 mmol). C_29_H_29_F_3_N_8_O_2_S_2_. *m*/*z* 642.73. LCMS [M + 1]^+^ = 643^+1^.

^1^H NMR spectrum (DMSO-*d*_6_, δ, ppm, *J*/Hz): 1.70–1.93 (m, 2H, CH_2_), 2.07–2.50 (m, 8H, 4CH_2_), 2.63 (d, ^3^J_HH_ = 13.83, 1H, CH_2_), 2.85 (d, ^3^J_HH_ = 12.96, 1H, CH_2_), 3.29–3.45 (m, 2H, CH_2_), 3.93–4.11 (m, 2H, CH_2_), 4.49–4.67 (m, 2H, CH_2_), 6.89–7.00 (m, 1H, aromatic), 7.08–7.20 (m, 1H, aromatic), 7.32–7.54 (m, 5H, aromatic), 7.65–7.80 (m, 1H, N=CH-N), 8.25–8.34 (m, 1H, N=CH-N), 8.46 (s, 1H, N=CH-N).

6-[3-[4-[2-(2,4-difluorophenyl)-2-hydroxy-3-(1H-1,2,4-triazol-1-yl)propyl]piperazin-1-yl]propyl]-3-(4-chlorophenyl)-2-thioxo-thiazolo[4,5-d]pyrimidin-7-one (III) was prepared as a light-yellow powder in 41% yield (0.27 g) from compound 3-(4-chlorophenyl)-2-methyl-6-(3-piperazin-1-ylpropyl)-2-thioxo-thiazolo[4,5-d]pyrimidin-7-one (R^1^ = Cl, R^2^ = H) (0.42 g, 0.1 mmol) and compound 1 (0.33 g, 0.10 mmol). C_29_H_29_ClF_2_N_8_O_2_S_2_ *m*/*z* 659.186; LCMS [M + 1]^+^ = 660.0^+1^. Yield 41%.

^1^H NMR spectrum (DMSO-*d*_6_, δ, ppm, *J*/Hz): 1.77–1.94 (m, 2H, CH_2_), 2.13–2.50 (m, 8H, 4CH_2_), 2.62 (d, ^3^J_HH_ = 14.00, 1H, CH_2_), 2.85 (d, ^3^J_HH_ = 14.67, 1H, CH_2_), 3.32–3.52 (m, 2H, CH_2_), 3.99 (t, ^3^J_HH_ = 6.27, 2H, CH_2_), 4.47–4.62 (m, 2H, CH_2_), 6.88–7.00 (m, 1H, aromatic), 7.06–7.19 (m, 1H, aromatic), 7.30–7.41 (m, 1H, aromatic), 7.42–7.50 (m, 2H, aromatic), 7.59–7.70 (m, 2H, aromatic), 7.73 (s, 1H, N=CH-N), 8.30 (s, 1H, N=CH-N), 8.47 (s, 1H, N=CH-N).

It should be noted that the obtained hybrids exist as racemic mixtures. This fact was confirmed by analysing the optical rotation of compounds I–III using WZZ-2B automatic polarimeter. The optical rotation is zero for all hybrids. Solid state characterization of synthesized compounds was carried out by methods PXRD and DSC (Appendix A, respectively). The DSC thermograms of all the hybrid samples were smooth and had no endothermic peaks within the interval from 25 to 220 °C of the heating cycle, which indicates that the samples had an amorphous structure. The results of the PXRD analysis for compounds I–III completely confirm the differential scanning calorimetry data: the diffractograms displayed a halo peak, which confirmed the lacking crystalline nature of the compounds.

### 3.3. Antifungal Activity Study

The in vitro antifungal activity of the target compounds was measured by means of the minimal inhibitory concentrations (MIC) with fluconazole as the control drug. Procedure was carried out by the microbroth dilution method in 96-well plates according to the recommendations of the Clinical and Laboratory Standards Institute [35]. The detailed information was reported previously [7]. The investigated pathogenic species included *C. parapsilosis ATCC 22019, C. utilis 84* and *C. glabrata 61.* The sensitivity was evaluated visually, after the incubation at 35 °C for 24 h and the value was compared with the growth density in the reference culture without the sample. Each test was conducted in triplicate.

### 3.4. Kinetic Solubility

Kinetic solubility was measured based on the shake-flask method in buffers pH 2.0 and 7.4 at 298.15 K for 75 h. First, the excess amount of the compound was added to respective buffer solution (volume 10 mL) in Pyrex glass tubes. Further, the tubes were shaking in the air thermostat at 25 °C. The amount of the dissolved sample was measured by taking aliquots of the media at pre-determined time points. The suspension was filtered by hydrophilic filter Millex-GV 0.22 µm (Ireland). The solubility, as the concentration of the filtrates, was quantified using a spectrophotometer Cary-50 (Varian, Palo Alto, CA, USA). The wavelengths corresponding to the absorption maximums for compounds studied in in selected buffer solutions have been specified as 332 nm. Each experiment was repeated in triplicate.

### 3.5. Equilibrium Solubility

Shake-flask method as “gold standard” was used for determination thermodynamic solubility synthesized compounds in buffer solutions (pH 2.0 and 7.4) and 1-octanol in the temperature range of 293.15–313.15 K with a step of 5 degrees at atmospheric pressure. Excessive amounts of compound were added to glass vials c 12 mL of selected solvent. The heterogeneous system is closed with a lid and vigorously stirred at the selected temperature in an air thermostat with a stirring device until the solubility equilibrium has been reached. The time required for establishing a constant value of the solution concentration was determined from the solubility kinetic dependences and averaged 2 days. The equilibrated solutions were first survived to aim sedimentation of solid phase during 10 h, then centrifuged at 20,000 rpm for 30 min at specific temperatures, and then quickly filtered using a syringe equipped with a hydrophilic filter Millex-GV 0.22 µm (Ireland). Concentration of saturated solution was measured using spectrophotometer Cary-50 (Varian, USA) with an accuracy of 2–4%. The absorbances of compounds studied were measured at wavelengths 332 and 334 nm in buffer solutions and 1-octanol, respectively. If necessary, an aliquot was diluted with an appropriate solvent. The final pH of the saturated buffers was monitored. The solubility of compounds in each solvent was determined by repeating the procedure three times.

Mole fraction concentration was calculated from solubility value *S* expressed in mol∙L^−1^ using Equation (10):(10)$$x=\frac{M_{2}S}{S(M_{2}-M_{1})+1000\rho },\;$$
where *M*_1_ and *M*_2_ are the molar masses of solute and solvent, respectively, and *ρ* (g∙cm^−3^) is the density of the pure solvents. The mole fraction solubilities for buffer solutions were calculated taking into account the buffer compositions. The used values of densities are given in Appendix A.

### 3.6. Partition Experiment

The octanol/buffer pH 7.4 partition coefficient log*P_O/B_* is measured in a static mode using the shake-flask method. The procedure has been presented as standard in the Organization for Economic Cooperation and Development (OECD) guidelines for testing chemicals [36]. The equipment described above was also used for the determination of the 1-octanol/water partition coefficients. The procedure was as follows: (1) solvents (buffers and 1-octanol pre-saturated by stirring at room temperature for 24 h; (2) 1-octanol saturated solution at a concentration of the solubility of the substance examined was prepared; (3) the solution was placed in a glass ampoule and an identical volume of buffer was added; (4) the ampoule was placed in a thermostat at fixed temperature; (5) the measurement was continued for 2 days with continuous stirring; (6) the initial and final concentrations in the aqueous and organic phases were determined by spectrophotometry. The 1-octanol/buffer partition coefficients *P_o/b_* were calculated as the ratio of equilibrium concentrations in the organic and aqueous phases: *P**_o_*/*_b_* = *s_o_/s_b_*(11)
where *s_o_* and *s_b_*—molar concentrations of compound in the 1-octanol and buffer phases, respectively. The correctness of the obtained value of each compound was verified by checking the mass balance of the starting amount of the compound and the total amount of the compound partitioned between the two phases.

To aim evaluated transfer (partition) thermodynamic functions, the 1-octanol/buffer partition coefficients *P^*^
_o/b_* as the ratio of equilibrium concentrations in the organic (*x_o_*) and aqueous phases (*x_b_*) expressed in mole fractions were calculated: *P* _o/b_* = *x_o_/x_b_*(12)

Standard change enthalpy of transfer (∆_*tr*_*H^o^*) was obtained by means of temperature dependence of partition coefficient using van’t Hoff method, assuming that ∆_tr_*H^o^* does not depend on temperature in the investigated range:(13)$$\frac{d(\ln P_{o/b}^{*})}{dT}=\frac{\Delta_{tr}H^{o}}{RT^{2}}$$

The standard change in Gibbs free energy of transfer (∆_*tr*_*G^o^*) from the buffer to the organic phase was calculated by Equation (14):∆_*tr*_*G*^*o*^ = −*RTlnP***_o_/_b_*(14)

The standard change entropy of transfer (∆_*tr*_*S^o^*) is determined as follows:∆_*tr*_*S*^*o*^ = (∆_*tr*_*H*^*o*^ − ∆_*tr*_*G*^*o*^)/*T*(15)

### 3.7. Theoretical Basis

#### 3.7.1. Van’t Hoff Equation

The van’t Hoff equation is widely used to quantitatively describe the solid–liquid equilibrium as the relationship between the experimental solubility and the temperature considering the influence of the solvent [37,38]. In this equation, the logarithm of mole fraction solubility *x* is linear with the reciprocal thermodynamic temperature of system [39]:(16)$$lnx=A+\frac{B}{\left(T/K\right)}\;$$
where *A* and *B* are the model constants calculated by the least square analysis.

#### 3.7.2. Modified Apelblat Equation

The modified Apelblat equation was first proposed by Apelblat [40] and successfully applied to correlate and determine the data on the solubility of various substances [41,42], expressed as follows: (17)$$lnx=A+\frac{B}{T/K}+Cln(T/K)\;$$
where *x* is the experimental saturated mole fraction solubility of the solute, *T* is the absolute temperature, *A*, *B,* and *C* are the empirical fitting parameters. Values *A* and *B* represent the variation in the solution behaviour, resulting from the non-ideality of the solute solubility, whereas value of *C* represents the association between the temperatures and the enthalpy of fusion.

#### 3.7.3. Evaluation of Precision for Used Models

To compare prediction errors of solubility data using thermodynamic models, accuracy parameters RAD (average relative deviation) and RMSD (root mean square deviation) are used. The relative average deviation (RAD) reveals how much each measurement differs, on average, from the arithmetic mean of the data set and is determined by the equation:(18)$$RAD=\frac{1}{N}\sum_{i=1}^{N}{\left|\frac{x_{exp}-x_{cal}}{x_{exp}}\right|}^{}$$
where *x_exp_*_-_ and *x_cal_* are the experimental and calculated values of solubility, *N* represents number of experimental points.

Root mean squared error (RMSD) is an excellent general-purpose error metric for numerical predictions and evaluate as the square root of the mean of the square of all of the error:(19)$$RMSD={\left|\frac{1}{N}\sum_{i=1}^{N}{\left(x_{exp}^{}-x_{cal}^{}\right)}^{2}\right|}^{1/2}$$

The designations in the equation are similar to the previous one.

## 4. Conclusions

The three novel bioactive hybrids based on fluconazole have been synthesized. Bioactive assay manifested that prepared compounds showed antifungal activities equal or greater than the reference drug fluconazole. The solid state of hybrids studied was characterized as amorphous via DSC and PXRD techniques. The equilibrium solubility of compounds was measured by the shake-flask method over a temperature range (298.15 to 313.15) K in the pharmaceutically relevant solvents: buffer solutions (pH 2.0 and 7.4) and 1-octanol. The hybrid compounds are poorly soluble in the buffer solution pH 7.4 and have similar solubility values changing at T = 298.15 K within the range (0.67–3.19) × 10^−4^ mol∙L^−1^. The solubility values in the hydrochloric buffer pH 2.0 are an order of magnitude higher, due to the ionization process of hybrids in acidic medium. Further, in 1-octanol, solubility of chemicals is maximum (0.75–1.23) × 10^−2^, which indicates the high lipophilicity of the substances and good permeation properties making their application in the form of ointments and creams favourable. The solubility behaviour of compounds predicted by Hansen solubility parameters adequately reflects the experimental results obtained in this work. The obtained solubility values were correlated by two thermodynamics models (modified Apelblat and van’t Hoff equations). Good agreement was observed between experimental and calculated solubility, while the modified Apelblat model shows the best correlation in all investigated solvents. Apparent partial molar parameters indicated that dissolution process was endothermic and enthalpy driven. The hydrophilic–lipophilic features of hybrids were reported based the temperature dependences partition coefficients in 1-octanol/buffer pH 7.4 system. The obtained values of log*P_o/b_* of hybrids were changed from 1.65 to 2.50, which exhibited their optimal biopharmaceutical properties: good balance between the solubility and permeability through passive diffusion. From the outcomes, it was determined that partition process of compounds studied from aqueous medium to lipid phase is driven by the entropy component of transfer Gibbs energy increasing the system disordering.

Thus, the experimental examination demonstrated that the new hybrids are promising candidates for further pharmaceutical trials as antifungal agents, since they exhibit a wide range of antifungal activity comparable to fluconazole and acceptable values of lipophilicity for oral absorption 1 < log*P_o/b_* < 3. The low solubility of the studied compounds is not, in our opinion, a critical factor, since it can be improved by creating water-soluble dosage forms using such modern approaches as solubilization, co-crystallization, micronization, emulsifications, etc.

## Figures and Tables

**Figure 1 molecules-27-06504-f001:**
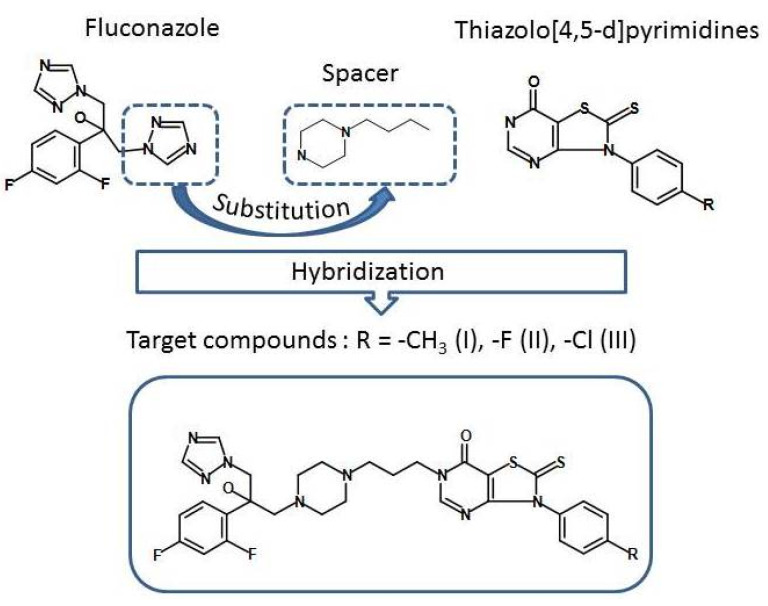
Design of novel target hybrids fluconazole and thiazolo[4,5-d]pyrimidine.

**Figure 2 molecules-27-06504-f002:**
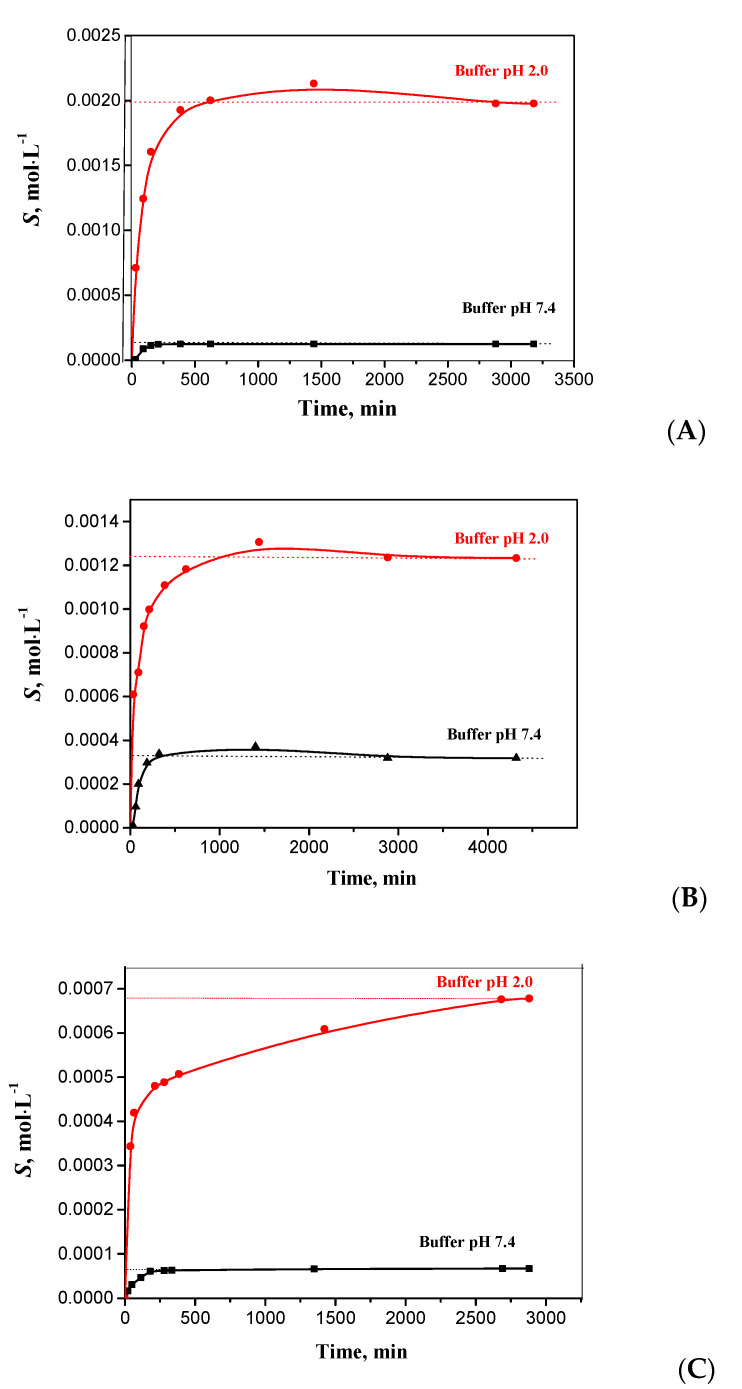
Solubility kinetic profiles of compounds I (**A**), II (**B**), and III (**C**) in buffer solutions (pH 2.0 and 7.4) at 298.15 K (dotted lines corresponds to equilibrium solubility).

**Figure 3 molecules-27-06504-f003:**
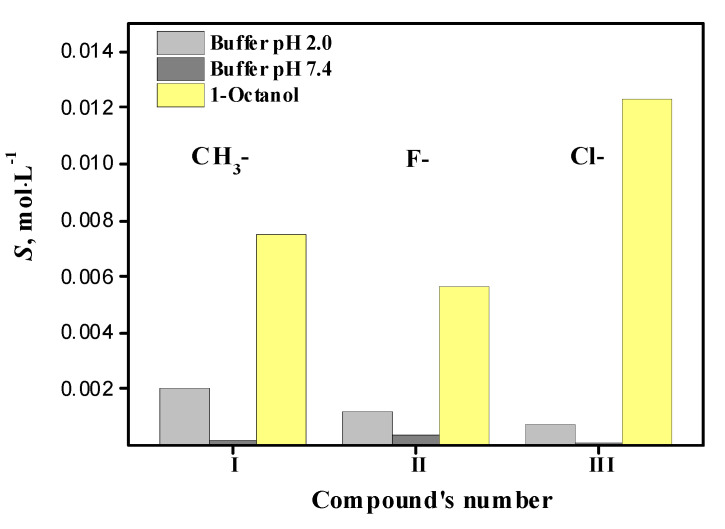
Solubility data of compounds I–III in selected solvents at 298.15 K.

**Figure 4 molecules-27-06504-f004:**
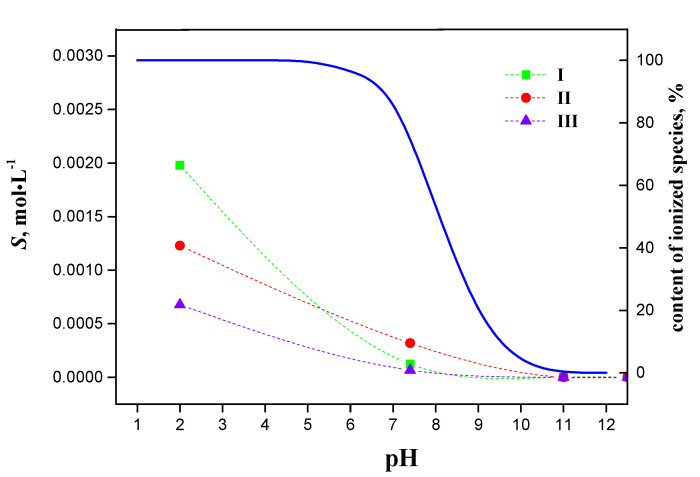
Characteristic ionization profile of compounds studied (blue line) and corresponding experimental solubility values at T = 298.15 K.

**Figure 5 molecules-27-06504-f005:**
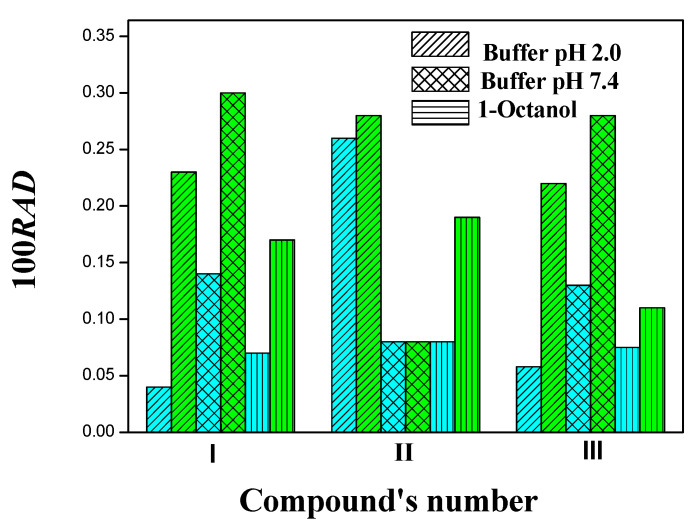
Comparative histogram of relative average deviations for used models: modified Apelblat equation—blue colour, van‘t Hoff equation—green colour.

**Figure 6 molecules-27-06504-f006:**
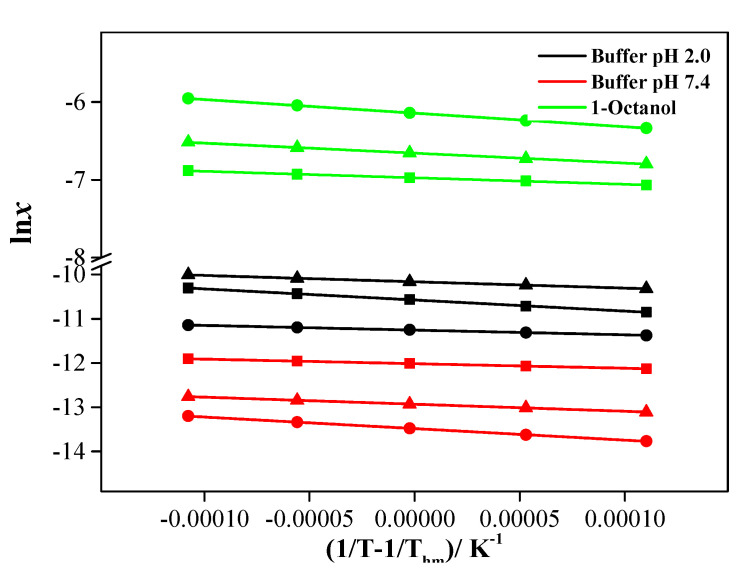
Van’t Hoff plot of lnx versus (1/*T* − 1/*T_hm_*) for compounds I–III in solvents used: ▲—I; ●—II; ■—III.

**Table 1 molecules-27-06504-t001:** Antifungal activity data of the hybrid compounds.

Compound	MIC (μg/mL)		
	*C. parapsilosis* ATCC 22019	*C. utilis* 84	*C. glabrata 61L*
I	0.5	4	2
II	0.1	1	4
III	0.25	2	2
Fluconazole	2	2	8

Relative errors are generally within 5%.

**Table 2 molecules-27-06504-t002:** Solubility (*S*, mol∙L^−1^ and *x*, mol. frac.) of compounds studied (amorphous state) in buffer solutions (pH 2.0 ^a^ and 7.4 ^b^) and 1-octanol at different temperatures and pressure *p* = 0.1 MPa.

T/K	I			II			III		
	*^a^* Buffer pH 2.0	*^b^* Buffer pH 7.4	1-Octanol	Buffer pH 2.0	Buffer pH 7.4	1-Octanol	Buffer pH 2.0	Buffer pH 7.4	1-Octanol
	*x* × 10^5^ (*S* × 10^3^)	*x* × 10^6^ (*S* × 10^4^)	*x* × 10^3^ (*S* × 10^3^)	*x* × 10^5^ (*S* × 10^3^)	*x* × 10^6^ (*S* × 10^4^)	*x* × 10^4^ (*S* × 10^3^)	*x* × 10^5^ (*S* × 10^4^)	*x* × 10^6^ (*S* × 10^5^)	*x* × 10^3^ (*S* × 10^2^)
293.15	3.3040 (1.8307)	2.0541 (1.1295)	1.1199 (7.0781)	1.9402 (1.0755)	5.4794 (3.0126)	8.5358 (5.3994)	1.1474 (6.3622)	1.0651 (5.8572)	1.7767 (1.1196)
298.15	3.5732 (1.9773)	2.2653 (1.2441)	1.1981 (7.5389)	2.2272 (1.2330)	5.8122 (3.1917)	9.0027 (5.6703)	1.2247 (6.7826)	1.2262 (6.7347)	1.9566 (1.2269)
303.15	3.8570 (2.1311)	2.4636 (1.3510)	1.2903 (8.0826)	2.5822 (1.4273)	6.1623 (3.3788)	9.3813 (5.8834)	1.3013 (7.1958)	1.4201 (7.7877)	2.1577 (1.3464)
308.15	4.1749 (2.3027)	2.6820 (1.4682)	1.3835 (8.6264)	2.9451 (1.6250)	6.4929 (3.5539)	9.8001 (6.1187)	1.3731 (7.5805)	1.6349 (8.9503)	2.3729 (1.4730)
313.15	4.5144 (2.4851)	2.9135 (1.5920)	1.4834 (9.2070)	3.3404 (1.8394)	6.8581 (3.7469)	10.2644 (6.3807)	1.4468 (7.9724)	1.8788 (10.2665)	2.5979 (1.6045)

*^a^* Composition of aqueous buffer pH 2.0: KCl (6.57 g in 1 L) and 0.1 mol·L^−1^ hydrochloric acid (119.0 mL in 1 L); *^b^* Composition of aqueous buffer pH 7.4: KH_2_PO_4_ (9.1 g in 1 L) and Na_2_HPO_4_·12H_2_O (23.6 g in 1 L). Standard uncertainties for mass salt (*m*) and volume of solution (V): *u*(*m*) = 5 mg, *u*(V) = 0.5 mL. Standard uncertainties are *u*(*T*) = 0.15 K, *u*(*p*) = 3 kPa, *u*(pH) = 0.02 pH units. Relative standard uncertainties for solubility in buffer solutions: *u_r_(S)* = 0.045 and *u_r_(x)* = 0.045. Relative standard uncertainties for solubility in 1-octanol: *u_r_(S)* = 0.04 and *u_r_*(*x*) = 0.04.

**Table 3 molecules-27-06504-t003:** Molar volumes and Hansen solubility parameters for compounds I–III and selected solvents.

Compound	*V*, cm^3^·mol^−1^	*δ_d_* MPa^0.5^	*δ_p_* MPa^0.5^	*δ_h_* MPa^0.5^	*δ_t_* MPa^0.5^	Δ*δ_t_* MPa^0.5^	∆*δ* MPa^0.5^
I	540.3	20.8	3.6	8.0	22.6	-	-
Buffer solutions	18.0	15.5	16.0	42.3	47.8	25.2	36.9
1-Octanol	157.7	17.0	3.3	11.9	21.0	1.6	5.4
II	524.8	21.0	4.3	8.9	23.2	-	-
Buffer solutions	18.0	15.5	16.0	42.3	47.8	24.6	35.8
1-Octanol	157.7	17.0	3.3	11.9	21.0	2.2	5.0
III	532.8	21.1	4.6	8.6	23.2	-	-
Buffer solutions	18.0	15.5	16.0	42.3	47.8	24.6	36.0
1-Octanol	157.7	17.0	3.3	11.9	21.0	2.2	5.4

**Table 4 molecules-27-06504-t004:** Apparent thermodynamic solubility functions of compounds studied in buffer solutions (pH 2.0 and 7.4) and 1-octanol at *T_hm_* = 302.98 K and pressure *p* = 0.1 MPa.

Compound	$\Delta G_{sol}^{o}\;\mathrm{kJ}\cdot {\mathrm{mol}}^{-1}$	$\Delta H_{sol}^{o}\;\mathrm{kJ}\cdot {\mathrm{mol}}^{-1}$	$T\Delta S_{sol}^{o}\;\mathrm{kJ}\cdot {\mathrm{mol}}^{-1}$	$\Delta S_{sol}^{o}\;J\cdot {\mathrm{mol}}^{-1}\cdot K^{-1}$
Buffer pH 2.0				
I	25.4 ± 0.5	11.9 ± 0.1	−13.5	−45.2 ± 2.4
II	26.6 ± 0.5	20.9 ± 0.2	−5.7	−19.1 ± 1.1
III	28.0 ± 0.4	8.8 ± 0.1	−19.2	−64.4 ± 3.8
	Buffer pH 7.4			
I	32.2 ± 0.6	13.2 ± 0.1	−19.0	−63.6 ± 4.1
II	29.9 ± 0.6	8.5 ± 0.1	−21.3	−71.4 ± 4.2
III	33.7 ± 0.7	21.7 ± 0.2	−12.0	−40.3 ± 2.3
	1-Octanol			
I	16.8 ± 0.2	10.6 ± 0.2	−6.2	−20.8 ± 1.1
II	17.4 ± 0.3	6.9 ± 0.1	−10.4	−35.1 ± 1.8
III	15.4 ± 0.3	14.5 ± 0.1	−0.9	−3.1 ± 0.2

The standard uncertainties are *u*(*T*) = 0.15 K and *u*(*p*) = 3 kPa.

**Table 5 molecules-27-06504-t005:** Experimental concentrations (*s_o_*, *s_b_*—mol∙L^−1^ and *x_o_*, *x_b_*—mole fraction in 1-octanol and buffer, respectively) and partition coefficients (log*P_o/b_*, log*P*^*^*_o/b_*—calculated by concentrations expressed in mol∙L^−1^ and mole fraction, respectively) for compounds studied in the system 1-octanol/buffer pH 7.4 at different temperatures and pressure *p* = 0.1 MPa.

***T*/K**	**I**						**II**					
	***s_b_* × 10^6^**	***s_o_* × 10^3^**	**log*P_o/b_***	***x**_b_* × 10^7^**	***x_o_* × 10^4^**	**log*P*^*^*_o/b_***	***s_b_* × 10^5^**	***s_o_* × 10^3^**	**log*P_o/b_***	***x**_b_* × 10^7^**	***x_o_* × 10^4^**	**log*P*^*^*_o/b_***
293.15	10.2	2.88	2.45	1.82	4.54	3.39	1.47	3.39	2.36	2.67	5.34	3.30
298.15	9.11	2.88	2.50	1.63	4.57	3.44	1.35	3.39	2.40	2.45	5.37	3.34
303.15	7.95	2.88	2.56	1.43	4.59	3.50	1.25	3.39	2.43	2.38	5.39	3.37
308.15	7.16	2.88	2.60	1.29	4.61	3.55	1.15	3.39	2.47	2.08	5.42	3.41
313.15	6.38	2.88	2.65	1.15	4.63	3.60	1.06	3.39	2.50	1.94	5.44	3.45
A *^a^*		6.71 ± 0.05							5.61 ± 0.05			
B *^a^*		972 ± 16							678 ± 16			
R *^b^*		0.9996							0.9983			
σ *^c^*		0.7 × 10^−4^							0.2 × 10^−4^			
***T*/K**	**III**											
	***s_b_* × 10^6^**	***s_o_* × 10^4^**	**log*P_o/b_***	***x_b_* × 10^7^**	***x_o_* × 10^5^**	**log*P*^*^*_o/b_***						
293.15	9.29	3.89	1.62	1.69	6.14	2.56						
298.15	8.77	3.90	1.65	1.60	6.17	2.59						
303.15	8.29	3.91	1.67	1.51	6.21	2.61						
308.15	7.80	3.91	1.70	1.42	6.24	2.64						
313.15	7.36	3.92	1.73	1.35	6.28	2.69						
A *^a^*		4.25 ± 0.02										
B *^a^*		495 ± 21										
R *^b^*		0.9972										
σ *^c^*		0.3 × 10^−4^										

*^a^* parameters of the correlation equation: log*P**_*o/b*_ = A + B/T; *^b^* R is the pair correlation coefficient; *^c^* σ is the standard deviation.

**Table 6 molecules-27-06504-t006:** Thermodynamics functions of transfer for compounds studied in system 1-octanol/buffer pH 7.4 at 298.15 K and p = 0.1 MPa. The standard uncertainties are u(T) = 0.15 K, u(p) = 3 kPa.

Compound	∆_tr_*G**^o^* kJ·mol^−1^	∆_tr_*H**^o^* kJ·mol^−1^	*T*∆_tr_*S**^o^* kJ·mol^−1^	∆_tr_*S**^o^* J·mol^−1^·K^−1^
I II III	−19.6 ± 0.5 −19.1 ± 0.3 −14.8 ± 0.2	18.3 ± 0.3 12.8 ± 0.2 9.5 ± 0.1	38.0 ± 0.9 31.9 ± 0.8 24.3 ± 1.0	127.4 ± 4.7 106.9 ± 2.9 81.5 ± 1.5

The standard uncertainties are *u*(*T*) = 0.15 K, *u*(*p*) = 3 kPa. The satisfaction of the inequality *T*∆_tr_*S^o^* > ∆_tr_*H^o^* for all the studied compounds means that the partition process for all the studied compounds from the aqueous medium to the organic phase is driven by the entropy component of the Gibbs energy of the transfer.

**Table 7 molecules-27-06504-t007:** Information about chemicals used in this study.

Chemical Name	CAS Register No.	Formula	M/g mol^−1^	Source	Mass Fraction Purity
6-[3-[4-[2-(2,4-difluorophenyl)-2-hydroxy-3-(1H-1,2,4-triazol-1-yl)propyl]-1-piperazinyl]propyl]-2,3-dihydro-3-(4-methylphenyl)-2-thioxo-thiazolo[4,5-d]pyrimidin-7(6H)-one (I)	2637523-56-7	C_30_H_32_F_2_N_8_O_2_S_2_	638.75	Synthesis	≥0.96
6-[3-[4-[2-(2,4-difluorophenyl)-2-hydroxy-3-(1H-1,2,4-triazol-1-yl)propyl]-1-piperazinyl]propyl]-2,3-dihydro-3-(4-fluorophenyl)-2-thioxo-thiazolo[4,5-d]pyrimidin-7(6H)-one (II)	2637523-57-8	C_29_H_29_F_3_N_8_O_2_S_2_	642.72	Synthesis	≥0.96
6-[3-[4-[2-(2,4-difluorophenyl)-2-hydroxy-3-(1H-1,2,4-triazol-1-yl)propyl]-1-piperazinyl]propyl]-2,3-dihydro-3-(4-chlorophenyl)-2-thioxo-thiazolo[4,5-d]pyrimidin-7(6H)-one (III)	2637523-58-9	C_29_ H_29_ClF_2_N_8_O_2_S_2_	659.17	Synthesis	≥0.96
1-Octanol	111-87-5	C_8_H_18_O	130.20	Sigma- Aldrich	≥0.99 ^a^
Potassium dihydrogen phosphate	7778-77-0	KH_2_PO_4_	136.08	Merck	≥0.99 ^a^
Disodium hydrogen phosphate dodecahydrate	10039-32-4	Na_2_HPO_4_·12H_2_O	358.14	Merck	≥0.99 ^a^
Potassium chloride	7447-40-7	KCl	74.55	Sigma- Aldrich	≥0.99 ^a^
Hydrochloric acid 0.1 mol/dm^3^ fixanal	7647-01-0	HCl	36.46	Sigma- Aldrich	≥0.99 ^a^
Fluconazole	86386-73-4	C_13_H_12_F_2_N_6_O	306.27	Quimica Sintetica	≥0.99 ^a^

^a^ As stated by the supplier.

## Data Availability

Not applicable.

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
