# Peer review of "Experimental Examination of Solubility and Lipophilicity as Pharmaceutically Relevant Points of Novel Bioactive Hybrid Compounds"

_molecules, 2022, doi:10.3390/molecules27196504_

Round 1
Reviewer 1 Report
This a well-conducted and well-documented study on physico-chemical properties of drug molecules. Authors have demonstrated the importance of such properties for the pharmaceutical significance of drugs. I have no criticism on the developed methodology and on the data presented in main text.
Otherwise, the broad audience readership of "Molecules" will miss some comments on the determined property values. What range of values would have been acceptable for the pharmaceutical significance of the drugs and to what extent do these molecules fulfil the expectations? I strongly recommend to insert a short discussion of these points at the end of the manuscript.
Finally, the supplementary information was not attached at the end of the manuscript and I did not find how to download it. Therefore, my reviewing is incomplete since I had no access to the experimental data presented in this part.
Author Response
Reply to Reviewer_1
Thank you for carefully consideration of our work and proposed improvement.
Comment:
What range of values would have been acceptable for the pharmaceutical significance of the drugs and to what extent do these molecules fulfil the expectations? I strongly recommend to insert a short discussion of these points at the end of the manuscript.
Reply:
We added in conclusion a brief discussion of the obtained properties of new hybrids as promising candidates for further pharmaceutical trials.

Reviewer 2 Report
· Replace 'radicals' with 'groups’ (Abstract: line 10; line 388). -CH3, -F and -Cl are substituents on the investigated substrates and they do not have a radical character.
· Replace first person with third person (lines: 86; 137; 160; 231; 260; 294; 447; 466; 487; 595; 692)
· Lines 430-437 provide the preparation of the two fluids used in the solubility studies. However, no reference is provided in this paragraph. Please provide the references used to prepare the simulated gastric fluid.
· Why was the hydrochloric buffer pH 2.0 (imitating the gastric juice medium) selected? Most Pharmacopeias (and most pharma solubility studies) are using pH 1.2 as the simulated gastric fluid and not pH 2.0. Please provide a reference that shows the importance of using pH 1.2 vs pH 2.0.
Author Response
Reply to Reviewer_2
Thank you for carefully consideration of our work and proposed improvement.
Comment 1:
Replace 'radicals' with 'groups’ (Abstract: line 10; line 388). -CH3, -F and -Cl are substituents on the investigated substrates and they do not have a radical character.
Reply:
We have been replaced 'radicals' with 'groups’ (Abstract: line 10; line 388).
Comment 2:
Replace first person with third person (lines: 86; 137; 160; 231; 260; 294; 447; 466; 487; 595; 692)
Reply:
We have been changed first person to third person in all cases.
Comment 3:
Lines 430-437 provide the preparation of the two fluids used in the solubility studies. However, no reference is provided in this paragraph. Please provide the references used to prepare the simulated gastric fluid.
Reply:
The reference to preparation buffer solution pH 2.0 (simulated gastric fluid) has been provided in Experimental part: Buffers. A Guide for the Preparation and Use of Buffers in Biological Systems. Mohan, C. (Ed.) EMD Bioscience: Darmstadt, Germany, 2006.
Comment 4:
Why was the hydrochloric buffer pH 2.0 (imitating the gastric juice medium) selected? Most Pharmacopeias (and most pharma solubility studies) are using pH 1.2 as the simulated gastric fluid and not pH 2.0. Please provide a reference that shows the importance of using pH 1.2 vs pH 2.0.
Reply:
Тhe pH is an especially important factor for dissolution. Аccording to the literature data [Kerns, E.H.; Di, L. Drug-like Properties: Concepts, Structure Design and Methods, Academic Press, 2008; D. Hörter, J.B. Dressman. Influence of physicochemical properties on dissolution of drugs in the gastrointestinal tract. Advanced Drug Delivery Reviews, Vol. 46, 1 2001, 75-87. https://doi.org/10.1016/S0169-409X(00)00130-7], the pH of a fasted stomach medium varies from 1.4 to 2.1. The pH value of the buffer solution used in our work refers to this interval. The needed reference has been provided in manuscript.
